# Unlocking the Power of Sankalpa in Yoga Nidra Practice: Cognitive Restructuring Processes and Suggestions for Athletes’ Health

**DOI:** 10.3390/healthcare13161957

**Published:** 2025-08-10

**Authors:** Selenia di Fronso

**Affiliations:** Department of Theoretical and Applied Sciences, eCampus University, 22060 Novedrate, CO, Italy; selenia.difronso@uniecampus.it

**Keywords:** yoga, sport, pratyāhāra, well-being, recovery

## Abstract

This opinion article aims to highlight the potential mechanisms behind a specific stage of Yoga Nidra (YN) practice, i.e., the formulation and repetition of Sankalpa, encouraging scholars to further study it and providing athletes with suggestions on how to use it for their sport and health. YN can be defined as a meditation practice encompassing a sequence of breathing, guided body awareness, visualization and cognitive restructuring process exercises. According to preliminary results, YN stimulates a hypnagogic state generally associated with improvements in sleep parameters, thus enhancing recovery and health in different populations including athletes. Cognitive restructuring processes can be stimulated by the formulation of Sankalpa, a YN element comparable to positive self-instructions used to counteract dysfunctional cognitions. From a practical standpoint, the formulation of Sankalpa involves expressing an intention that could positively influence body, mind and emotions. For example, Sankalpa might stop or reverse unhealthy thought patterns, resulting in greater mental health. It might also foster intrinsic motivation and enhance emotional intelligence by strengthening mental resilience. In particular, athletes could use Sankalpa as an affirmation to awaken any strength they may feel is necessary to provide them with stress–recovery balance and mental health. However, additional research on this topic is needed to better elucidate Sankalpa’s mechanisms/effects and better integrate its formulation into sport programs.

## 1. Introduction

Mental and physical recovery in competitive sports is one of the most crucial determinants of performance success and health [1]. To aid recovery and health, athletes should improve their recovery skills, adopting strategies that positively influence their autonomic nervous system and muscle tension. Relaxation strategies, such as mindfulness-based meditations, can be used for this purpose [2]. Also, various yoga strategies often induce several physical and/or mental changes referred to as the relaxation response [3]. Among these strategies, Yoga Nidra (YN; literally, yogic sleep) is gaining increasing attention, in both the general and the athletic population. Overall, YN can be described as a promising non-invasive therapy or adjuvant for different concerns [4] that induces a specific mental condition between wakefulness and sleep. In this condition, the individual is physiologically asleep yet is able to follow the instructions coming from the guide [5].

This practice is generally associated with significant improvements in sleep parameters such as sleep onset latency (the time it takes from full wakefulness to the first stage of sleep) and sleep quality [6,7]. Moreover, despite limited evidence, it seems to be effective in post-traumatic stress disorder and psychological well-being [8]. Additionally, preliminary results of a YN intervention demonstrated improvements in sleep impairment in older adults compared to the control group and promising trends in depression symptoms and pain severity [9]. A short YN intervention recently reduced depression and salivary cortisol levels in a large sample when compared to a waitlist control group and to an active control group listening to music [10]. In a recent study, YN might have contributed to enhancing both the recovery and sleep quality of two karate athletes, inducing sport-specific effects in the male athlete (e.g., self-efficacy enhancement) and emotional and physical effects in the female athlete (e.g., decreases in perceived stress and somatic pre-sleep arousal) [11]. Of note, these findings derive from a case study, but this was based on an idiosyncratic approach and on the notion that each individual evaluates their recovery differently and has different levels of stress [12]. Also, a YN program recently enhanced self-confidence and decreased anxiety among female athletes [13].

Overall, YN might lead to restoration and rejuvenation because of a parasympathetic dominance, which increases slow-wave sleep and improves both subjective and objective sleep quality [14]. However, it might systematically benefit mind, body and recovery through different pathways [8]. By relying on this framework, this opinion article aims to shed light on the potential mechanisms behind the formulation of Sankalpa, one of the structured and organized stages of YN, encouraging scholars to further study it and providing athletes with suggestions on how to use it for their sport and, in particular, for their health.

YN, executed in the supine position, is composed of a sequence of guided body awareness, visualization, breathing and cognitive restructuring process exercises [7,15]. For an accurate overview of the different stages of a YN session, see the article written by di Fronso and Bertollo in 2021 [16] and Figure 1. Precisely, this kind of yoga can be considered a ‘pratyāhāra’ (i.e., withdrawal of senses) practice, which solicits the disengagement of mind and mental activities from perceptual awareness [3]. This might not only translate into profound relaxation and stress reduction but might also initiate cognitive restructuring [17], helping people/athletes deconstruct unhelpful thoughts and rebuild them in a more balanced manner. Also, cognitive restructuring processes might help regulate negative emotions that usually occur after sport performance, promoting emotional detachment [18]. It is indeed extremely important for athletes to distance themselves from post-performance emotions, like anger, which can hinder recovery and increase arousal, leading to energy depletion [19]. One of the major elements at the base of cognitive restructuring is called Sankalpa, a YN element comparable to positive self-instructions used in cognitive behavioral therapy settings to counteract dysfunctional cognitions [7]. While the mechanisms and effects of other exercises practiced during YN, like breathing or body awareness exercises, are documented [20], to the best of my knowledge, this is the first article that solicits new research lines focused on Sankalpa (and athletes) and tries to “unlock its power”.

## 2. Meaning, Origin and Potential Mechanisms of Sankalpa

The stage of YN characterized by the formulation of Sankalpa involves expressing an intention, generated by introspection and deep reflection, that is expected to impact body, mind and emotions. From a philosophical perspective, Sankalpa is conceived as a deep, present-rooted statement that reflects an individual’s true self and innermost aspirations. Accordingly, this intention might direct actions and thoughts, likely allowing people to align with their life purposes and manifest their potential.

The Sanskrit origin of Sankalpa comes from San—i.e., a connection with the highest truth—and Kalpa, i.e., “vow”. Thus, it denotes an affirming resolve to do or achieve something. From a practical point of view, during this stage, the individual mentally affirms a personal resolution, an intention that should be short, clear and positive. Formulating the Sankalpa in positive and present terms and using simple words could make the intention more vivid and rooted in the mind. Timing is an important aspect when formulating Sankalpa. The ideal moment could be when the mind is most receptive and calm, such as at the beginning or the end of a YN practice. This is the reason why Sankalpa is formulated twice throughout a YN session.

As previously mentioned, this intention is generally dedicated to some goal of self-realization or personal improvement. The intention could be anything as simple as breaking a personal bad habit, like “I resolve to stop excessive screen time before bed,” or it could be something far more profound, like “I resolve to be more mindful”. Precisely, while the individual mentally recites the selected Sankalpa three times, passivity is the term used by YN masters to describe the mental separation that results from this resolve. In this passive condition, the self and the experiences that would typically elicit strong emotions are separated. Consequently, under these circumstances, the resolution could be more rapidly assimilated into the unconscious [3].

As stated earlier, cognitive restructuring processes induced by Sankalpa might stop or reverse unhealthy thought patterns, resulting in personal accomplishment and greater mental health [16,21]. However, Sankalpa could also be considered remarkable motivational fuel. This motivation is essential to reach a state of “flow”, in which individuals are completely immersed in the activity they are performing, experiencing a sense of thorough satisfaction and fulfillment. This state is also associated with greater creativity, learning and psychological well-being [4], all aspects that could be enhanced through the regular practice of YN and Sankalpa. This Sanskrit concept could thus have the capacity to direct people toward their objectives with lucidity and tenacity [22]. Generally, Sankalpa differs from goal-setting because it is not externally driven but deeply committed to personal growth [22]. It might foster intrinsic motivation but could also enhance emotional intelligence by providing the mental resilience needed to overcome obstacles and manage emotions when facing challenges [10,22]. In turn, acknowledging and dealing effectively with emotions may strengthen people’s health and well-being [23]. The repetition of positive affirmations could also be seen as a reactive kind of self-regulation through which the mind learns to form new habits and patterns [24].

## 3. Sankalpa for Sport and Health

In the context of sports, as mentioned above, YN and Sankalpa might have recently helped to increase karate athletes’ self-efficacy ratings, which are viewed as a relevant sport-specific aspect of recovery [11]. Overall, athletes, always taking into account their personal aspirations, could use Sankalpa as an affirmation to overcome any weakness affecting their body and sport performance (and life in general), and to awaken any other strength they may feel is necessary to provide them with stress–recovery balance and mental health.

Immediately before or during competitions, mentally reciting something like “I am developing/I develop…” or “I am expressing my sport potential/I express,” instead of “I am going to develop…” or “I am going to express..”, athletes could better create a Sankalpa that suits their needs. Other examples, especially when YN is executed after training and/or before competitions, could be “I am calm” or “I am successful”. A crucial suggestion is being one step ahead of desired outcomes, at the beginning and the end of the YN practice. However, being careful not to simply create rules, doubts and confusion, and staying congruent between what is meant to be achieved and the actual feeling state, the phrasing of Sankalpa can be in either the present or the future tense. In this case, the word “will” suggests “I will do it”, I use my willpower and I can accomplish anything if I set my mind to it. Understanding that goals can be accomplished is more significant than the exact words used for the formulation of Sankalpa.

Furthermore, athletes should be acknowledged that Sankalpa might be more effective when practiced within YN sessions due to calmness and receptiveness of the mind. Also, although there are no time limits, a few minutes dedicated to silently repeating Sankalpa within a longer practice could be sufficient. For example, during a YN session of about 30 min, 5 min might be dedicated to Sankalpa. Notably, a consistent practice is key. Of note, YN should not be practiced too close to competitions to avoid excessive lowering of arousal levels due to the activation of the parasympathetic system [25]. Thus, the moments before competitions, reciting Sankalpa might be independent from the meditative practice. On the other hand, the day before competitions, or after training/competitions, athletes, according to their time constraints, can have a YN session where the intention is normally repeated at the beginning and at the end of the practice.

In the early stages of YN practice, Sankalpa could act as a focal point for mental energies, thereby concentrating and directing the mind towards a specific intention or goal. As the practice continues, the Sankalpa is repeated and this is believed to increase the energy necessary for manifesting intentions and accomplishing goals (see also Figure 1). In this way, Sankalpa could enable control over one’s own cognitive (e.g., feeling anxious) and physiological states (e.g., increased heart rate, muscle tension), but it could also strengthen awareness skills and create the space for inner harmony. The introspection and the conscious creation of a Sankalpa could lead athletes to gain deeper insights into their values, aspirations and needs (e.g., recovery needs), reinforcing their self-awareness. The more athletes become aware of their (recovery) needs, the higher the satisfaction and contentment that underlie physical and mental health. During the first YN sessions, a vital suggestion to reach and maintain a final/greater body–mind recovery could be phrasing a simple or mini Sankalpa like “I am going to be focused’, or “I will get through my YN practice without sleeping”.

Sankalpa-based self-affirmations’ capacity to reduce the impact of negative emotions could be viewed as one explanation for their effectiveness [26]. Indeed, counteracting negative or dysfunctional emotions could help athletes focus on psychosocial resources like optimism, social support, self-worth and a sense of mastery. These are thought to decrease reaction to threats and defend athletes’ psychological health in general [26,27]. Moreover, emotional detachment [18] might be triggered by Sankalpa’s post-performance regulation of negative emotions. This “switch-off” is necessary for athletes to mitigate the negative impacts of high sport demands on their health and well-being [28]. Overall, affirmations like Sankalpa might decrease stress by redirecting attention towards positive intensions, improve well-being and (sport) performance, and make people in general, and athletes in particular, prone to behavioral changes [29].

## 4. Conclusions

Given the preliminary nature of YN research and the limited number of studies on the topic, the fact that the mechanism/effects of Sankalpa herein described are merely possibilities should be noted. On the other hand, grounded in the notion that Sankalpa can be analogous to positive self-instructions used to counteract dysfunctional cognition, it could be argued that establishing it might serve to initiate a process of change inside the subconscious mind [4]. Notably, YN practice and Sankalpa characteristics might positively affect athletes’ stress–recovery balance and general well-being [11]. In particular, Sankalpa and its effects might ultimately be viewed as a further and complementary explanation of YN-based mental and physical recovery enhancement. On the other hand, cognitive restructuring processes and emotional detachment might contribute to ameliorating athletes’ performance success, sleep quality [6] and recovery processes. However, while practical suggestions for athletes (and coaches) approaching YN practice are important, as well as highlighting preliminary study results concerning YN (and sport), and unlocking the power of Sankalpa, additional research on this topic is needed. Specifically, research should envisage protocols that better analyze Sankalpa’s effects/mechanisms, also from a qualitative and neurophysiological point of view. From a qualitative perspective, it could be useful to interview athletes about the effectiveness they attributed to Sankalpa while practicing YN. From a neurophysiological standpoint, intriguing insights could be obtained from studying, for example, brainwave behavior or cortical patterns linked to the repetition of Sankalpa in the early and last stages of YN practice. Moreover, key benefits of Sankalpa could be better established through the utilization of protocols including or omitting this stage in the YN practice, paving the way for advancing our understanding of the mechanisms behind such a comprehensive yoga technique. This would also allow Sankalpa (and YN) to be better integrated into sport preparation programs.

## Figures and Tables

**Figure 1 healthcare-13-01957-f001:**
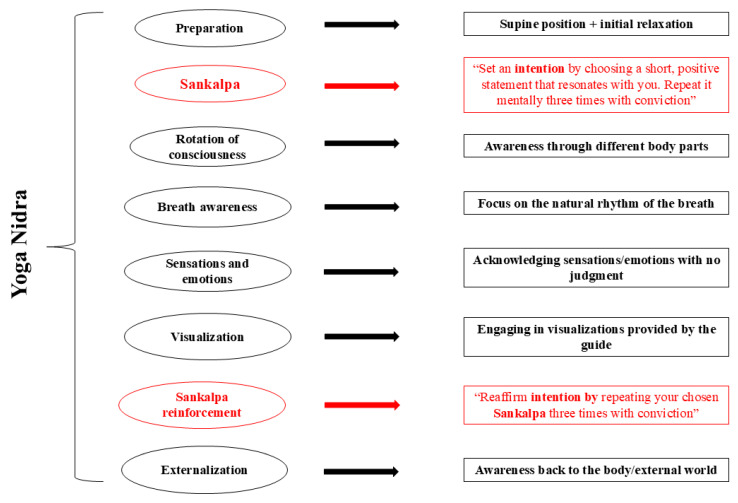
The stages of Yoga Nidra practice with a focus on Sankalpa.

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
