# Peer review of "Unlocking the Power of Sankalpa in Yoga Nidra Practice: Cognitive Restructuring Processes and Suggestions for Athletes’ Health"

_healthcare, 2025, doi:10.3390/healthcare13161957_

Round 1
Reviewer 1 Report
Comments and Suggestions for Authors
This opinion piece presents a compelling, well-articulated argument for the potential benefits of the Sankalpa stage in Yoga Nidra (YN) practice, with a specific focus on its applications in sport and athlete health. The manuscript is clearly structured, flows logically, and engages the reader with theoretical and practical perspectives on Sankalpa.
Strengths:
- The manuscript provides a unique and original viewpoint on Sankalpa, contextualizing it in the field of sports psychology and well-being.
- It references a wide range of contemporary studies, with adequate support for claims.
- The suggestions for athletes are practical and accessible, enhancing the translational value of the article.
Here my suggestions for Improvement:
- Clarify the article type earlier: While it becomes evident that this is an opinion article, stating this clearly in the introduction or abstract would help the reader.
- Minor edits for clarity:
- Line 16: consider replacing “might have an impact” with “can positively influence”.
- Line 136–137: the phrase “The feeling and the understanding…” could be revised to improve clarity.
- Consistency in referencing: Please check consistency in the formatting of references (e.g., punctuation and spacing in references 3, 4, and 14).
- Abstract formatting: The abstract reads more like an introduction. A clearer structure (objective → approach → key insights → implications) would enhance its effectiveness.
- Consider visual elements: A simple diagram showing the Sankalpa formulation process and its effects could increase readability and engagement, especially for practitioners.
Overall, the article adds value to the field and opens avenues for future research into Sankalpa’s mechanisms in cognitive and emotional domains.
Author Response
Dear Reviewer 1,
I appreciate the issues raised. I believe your comments were very helpful in improving the quality of my manuscript. Below, I responded to your queries point by point. To facilitate your work, any change was highlighted using yellow colour throughout the manuscript.
Comment 1: Clarify the article type earlier: While it becomes evident that this is an opinion article, stating this clearly in the introduction or abstract would help the reader.
Response 1: I agree that the article type should be clarified earlier, especially in the abstract. Accordingly, I started the abstract with the sentence: “This opinion article aims to highlight the potential mechanisms behind a specific stage of Yoga Nidra (YN) practice, i.e. the formulation and repetition of Sankalpa, encouraging scholars to further study it and providing athletes with suggestions on how to use it for their sport and health”. On the other hand, I left it where it was in the introduction as it generally reflects the standard position for the aim in any kind of article.
Comment 2: Minor edits for clarity
Line 16: consider replacing “might have an impact” with “can positively influence”.
Line 136–137: the phrase “The feeling and the understanding…” could be revised to improve clarity.
Response 2: “might have an impact” was revised accordingly. I apologize for being unclear with the phrase at lines 136-137. The sentence has now been changed as follows: “Understanding that goals could be accomplished is more significant than the exact words used for the formulation of Sankalpa”
Comment 3: Consistency in referencing: Please check consistency in the formatting of references (e.g., punctuation and spacing in references 3, 4, and 14)
Response 3: Thank you for highlighting inconsistency in the formatting of references. I have now checked and amended it.
Comment 4: Abstract formatting: The abstract reads more like an introduction. A clearer structure (objective → approach → key insights → implications) would enhance its effectiveness.
Response 4: Abstract changed accordingly. Thank you for the suggestion. It now reads as follows: “This opinion article aims to highlight the potential mechanisms behind a specific stage of Yoga Nidra (YN) practice, i.e. the formulation and repetition of Sankalpa, encouraging scholars to further study it and providing athletes with suggestions on how to use it for their sport and health. YN can be defined as a meditation practice encompassing a sequence of breathing, guided body awareness, visualization and cognitive restructuring process exercises. According to preliminary results, YN stimulates a hypnagogic state generally associated with improvements in sleep parameters, thus enhancing recovery and health in different populations including athletes. Cognitive restructuring processes can be stimulated by the formulation of Sankalpa, a YN element comparable to positive self-instructions used to counteract dysfunctional cognitions. From a practical standpoint, the formulation of Sankalpa involves expressing an intention that could positively influence body, mind and emotions. For example, Sankalpa might stop or reverse unhealthy thought patterns, resulting in greater mental health. It might also foster intrinsic motivation and enhance emotional intelligence by strengthening mental resilience. In particular, athletes could use Sankalpa as an affirmation to awaken any strength they may feel is necessary to provide them with stress-recovery balance and mental health. However, additional research on this topic is needed to better elucidate Sankalpa mechanisms/effects and better integrate its formulation into sport programmes.”
Comments 5: Consider visual elements: A simple diagram showing the Sankalpa formulation process and its effects could increase readability and engagement, especially for practitioners.
Response 5: Thank you for this important suggestion. I have now added a figure in the text called Figure 1.
Comment: Overall, the article adds value to the field and opens avenues for future research into Sankalpa’s mechanisms in cognitive and emotional domains.
Reply: Thank you once again for taking time to review my manuscript. I hope my amendments will meet your approval.
Reviewer 2 Report
Comments and Suggestions for Authors
Reviewer's comments
Abstract
- It could benefit from a more defined “Objective-Method-Conclusion” type structure, especially for interdisciplinary readers.
- Although this is an opinion article, it would be useful to note in the abstract that the results mentioned are preliminary or based on pilot or case studies.
Introduction
- Some paragraphs combine too many ideas (scientific evidence + philosophy + case experiences) without clear transition.
- Phrases such as “YN can be described as a mental health booster” should be qualified according to the level of evidence available.
- Could the author better delineate the focus between physical recovery and cognitive change in athletes
- Was consideration given to including a taxonomy of the types of Sankalpa used in sports contexts?
Meaning, origin and potential mechanisms of Sankalpa
- Include more references of studies that have operationalized Sankalpa in clinical or sports interventions.
Sankalpa for Sport and Health
- The term “autosuggestion” may be confused with pseudoscience; it is recommended to contrast it with psychological literature.
- Some phrases such as “mental energy accumulation” are vague and could benefit from greater conceptual or empirical precision.
- What strategies are suggested to evaluate whether a Sankalpa has been effective?
Conclusion
- The statement “Sankalpa may serve to initiate a process of profound change” should be accompanied by references or conditionality.
- It would be useful to specify which outcome measures are considered relevant (sleep, performance, cortisol, etc.).
- Does the author consider that Sankalpa can be easily integrated into existing sports psychology preparation programs?
It is recommended to improve the delimitation between theoretical speculation and empirical evidence, to clarify key terminology (autosuggestion, psychic energy), and to propose more operative future protocols to evaluate the effect of Sankalpa in sports contexts.
Author Response
Dear Reviewer 2,
Comment 1: Abstract -It could benefit from a more defined “Objective-Method-Conclusion” type structure, especially for interdisciplinary readers.
Although this is an opinion article, it would be useful to note in the abstract that the results mentioned are preliminary or based on pilot or case studies.
Response 1: Dear Reviewer, I appreciate the issues raised. I believe your comments were very helpful in improving the quality of my manuscript. Below, I responded to your queries point by point. To facilitate your work, any change was highlighted using yellow colour throughout the manuscript.
I agree with your comment about the abstract. It was modified considering both your comment and that of REV1. However, given this is an opinion article, method section was not included. Moreover, in the abstract I specified that the results are preliminary. The abstract reads now as follows: “This opinion article aims to highlight the potential mechanisms behind a specific stage of Yoga Nidra (YN) practice, i.e. the formulation and repetition of Sankalpa, encouraging scholars to further study it and providing athletes with suggestions on how to use it for their sport and health. YN can be defined as a meditation practice encompassing a sequence of breathing, guided body awareness, visualization and cognitive restructuring process exercises. According to preliminary results, YN stimulates a hypnagogic state generally associated with improvements in sleep parameters, thus enhancing recovery and health in different populations including athletes. Cognitive restructuring processes can be stimulated by the formulation of Sankalpa, a YN element comparable to positive self-instructions used to counteract dysfunctional cognitions. From a practical standpoint, the formulation of Sankalpa involves expressing an intention that could positively influence body, mind and emotions. For example, Sankalpa might stop or reverse unhealthy thought patterns, resulting in greater mental health. It might also foster intrinsic motivation and enhance emotional intelligence by strengthening mental resilience. In particular, athletes could use Sankalpa as an affirmation to awaken any strength they may feel is necessary to provide them with stress-recovery balance and mental health. However, additional research on this topic is needed to better elucidate Sankalpa mechanisms/effects and better integrate its formulation into sport programmes”.
Comment 2: Introduction - Some paragraphs combine too many ideas (scientific evidence + philosophy + case experiences) without clear transition. Phrases such as “YN can be described as a mental health booster” should be qualified according to the level of evidence available. Could the author better delineate the focus between physical recovery and cognitive change in athletes. Was consideration given to including a taxonomy of the types of Sankalpa used in sports contexts?
Response 2: Thank you for highlighting these points. I hope the way I revised the manuscript positively impacted also the transition of the ideas. Otherwise, I am open to other suggestions on how to improve this issue. Of note, literature about yoga tries to substantiate yoga philosophy by explaining/explicating scientific evidence. I changed the expression mental health booster as you suggested. The sentences reads now as follows: “YN can be described as a promising non-invasive therapy or adjuvant for different concerns that induces a specific mental condition between wakefulness and sleep”. The relationship between physical recovery and cognitive change in athletes has now been reinforced by adding the sentence “ Also, cognitive restructuring processes can help regulate negative emotions that usually occur after sport performances, promoting emotional detachment [18]. It is indeed extremely important for athletes to distance themselves from post-performance emotions, like anger, which can hinder recovery and increase arousal, leading to energy depletion [19].” towards the end of the introduction section (lines 73-77). Regarding your last question, as affirmed also in the manuscript, creating rules for Sankalpa can be counterproductive, athletes should be aware of their needs and formulate Sankalpa according to them; of course example can be found when I wrote “Mentally reciting something like "I am developing/I develop..." or "I am expressing my sport potential/I express," instead of, "I am going to develop..." or “I am going to express..” athletes could better create a Sankalpa that suits their needs. Other examples, especially when YN is executed after training and/or before a competition, could be “I am calm”, “I am successful”. This part has now been highlighted using yellow color. In this regard see also Figure 1 that was created following REV 1’ s suggestion.
Comment 3: Meaning, origin and potential mechanisms of Sankalpa - Include more references of studies that have operationalized Sankalpa in clinical or sports interventions.
Response 3: Thank you for this suggestion. I understand your reasoning about the inclusion of studies that have operationalized Sankalpa in clinical or sport interventions. However, all the studies relevant to this opinion that have adopted Yoga Nidra and consequently Sankalpa, have been included in the introduction (see for example ref 9 and 11). In this paragraph I tried to focus my attention on the potential mechanisms of Sankalpa based on the notion that it can be comparable to positive self-instructions used in cognitive behavioral therapy settings to counteract dysfunctional cognitions and facilitate cognitive restructuring. The intention, as stated in the aim, is to encourage scholars to further explore YN especially to deepen the role of Sankalpa.
Comment 4: Sankalpa for Sport and Health - The term “autosuggestion” may be confused with pseudoscience; it is recommended to contrast it with psychological literature. Some phrases such as “mental energy accumulation” are vague and could benefit from greater conceptual or empirical precision. What strategies are suggested to evaluate whether a Sankalpa has been effective?
Response 4: Thank you for these important comments and suggestions about the vagueness of the sentence and term “autosuggestion”. Overall, I modified the period as follows “ In the early stages of YN practice, Sankalpa could act as a focal point for mental energies thereby concentrating and directing the mind towards a specific intention or goal. As the practice continues, the Sankalpa is repeated and this is believed to increase the energy necessary for manifesting intentions and accomplishing goals. In this way, Sankalpa could enable control over one’s own cognitive (e.g., feeling anxious) and physiological states (e.g., increased heart rate, muscle tension) but it could also strengthen awareness skills and create the space for inner harmony. The introspection and the conscious creation of a Sankalpa could lead athletes gain deeper insights into their values, aspirations, and needs (e.g., recovery needs), reinforcing their self-awareness…”. On the other hand, the response to your last question was incorporated into conclusion section. For example, evaluation of the effectiveness of Sankalpa can be a specific part of qualitative studies aimed at interviewing athletes about the role of Sankalpa on the results obtained through YN practice. Moreover, as written in the conclusion section, “key benefits of Sankalpa could be better established through the utilization of protocols including or not this stage in the YN practice, paving the way to advance our understanding of mechanisms behind such a comprehensive yoga technique”.
Comment 5: The statement “Sankalpa may serve to initiate a process of profound change” should be accompanied by references or conditionality
It would be useful to specify which outcome measures are considered relevant (sleep, performance, cortisol, etc.).
Does the author consider that Sankalpa can be easily integrated into existing sports psychology preparation programs?
Response 5: I appreciate the issues raised. Please, consider that I used conditionality and ref (n 4) but I have now slightly toned down the sentence deleting “profound” to be more neutral. On the other hand, outcomes measures were specified stating…”cognitive restructuring processes and emotional detachment may contribute to ameliorating athletes’ performance success, sleep quality [6], and recovery processes”. I also added the following sentence “From a qualitative perspective, it could be useful interviewing athletes about the effectiveness they attributed to Sankalpa while practicing YN. From a neurophysiological stand-point, intriguing insights could be obtained studying, for example, brainwaves behavior or cortical patterns linked to the repetition of Sankalpa in the early and last stages of YN practice.”. With regards to the last point raised, I added the following sentence: … “This would also allow to better integrate Sankalpa (and YN) into sport preparation programmes”, based on the notion that accurate protocols about the utilization of Sankalpa are related to more scientific evidence needed.
Comment 6: It is recommended to improve the delimitation between theoretical speculation and empirical evidence, to clarify key terminology (autosuggestion, psychic energy), and to propose more operative future protocols to evaluate the effect of Sankalpa in sports contexts.
Response 6: Please see my previous response regarding protocols. Thank you once again for taking the time to review my manuscript. I hope my amendments (including delimitation between theoretical speculation and theoretical evidence, more clarity on key terminology etc.) will meet your approval.
Reviewer 3 Report
Comments and Suggestions for Authors
This article discusses the theoretical and practical aspects of how the concept of Sankalpa in Yoga Nidra practice can support mental health, recovery and motivation in athletes through cognitive restructuring processes. The author discusses the potential benefits of this ancient meditation practice, which is gaining increasing attention in the field of sport sciences, both in relation to the conceptual framework and previous limited studies. This evaluation is based on the scientific validity, originality, methodological adequacy and contribution to the literature.
The abstract gives an overview of the purpose, the importance of the topic and the applications. However, it should be made clearer that it is an opinion piece and the language of findings should be avoided (e.g. phrases such as “first study results” should be presented more cautiously).
Introduction
The need for recovery and mental health in athletes is clearly presented.
Very up-to-date and comprehensive, with references to Yoga Nidra studies published in the last 3 years.
The focus on the Sankalpa component within Yoga Nidra makes this article unique. However, how this contribution fills a gap in the literature could be more strongly emphasized.
Methodology
The article is based on opinion and theoretical explanation, not empirical. The format is appropriate in this context.
This section is not valid as it does not include analysis based on findings. However, the author has carefully interpreted the results of previous studies.
Findings
The potential impacts of Sankalpa are explained in a structured way.
Some suggestions are supported by examples (e.g. “I am calm”, “I express my sport potential”). However, the practical validity of the suggestions should be examined more critically. The reader should not be left with the question of what these suggestions are based on.
Discussion
Consistent and comprehensive with the relevant literature. However, the methodological limitations of research on the effects of Yoga Nidra and Sankalpa should be given more space.
The limitations of non-evidence-based recommendations should be made clear, especially given the opinion piece format.
There are some very good guidance (recommendations for qualitative/neurophysiological analysis, etc.).
The potential of Sankalpa for mental healing and motivational boosting is well summarized.
Conclusion
Concise and clear. However, in the conclusion, more cautious language (e.g. “Sankalpa may...”) should be preferred, in line with the nature of the opinion piece.
Author Response
Dear Reviewer 3,
Comment: This article discusses the theoretical and practical aspects of how the concept of Sankalpa in Yoga Nidra practice can support mental health, recovery and motivation in athletes through cognitive restructuring processes. The author discusses the potential benefits of this ancient meditation practice, which is gaining increasing attention in the field of sport sciences, both in relation to the conceptual framework and previous limited studies. This evaluation is based on the scientific validity, originality, methodological adequacy and contribution to the literature.
Reply: I thank the Reviewer for the positive feedback on my manuscript.
Comment 1: Abstract - The abstract gives an overview of the purpose, the importance of the topic and the applications. However, it should be made clearer that it is an opinion piece and the language of findings should be avoided (e.g. phrases such as “first study results” should be presented more cautiously).
Response 1: Dear Reviewer, I appreciate the issues raised. I believe your comments were very helpful in improving the quality of my manuscript. Below, I responded to your queries point by point. To facilitate your work, any change was highlighted using yellow colour throughout the manuscript. In order to made clearer that this is an opinion paper I started the abstract as follows: “This opinion article aims to highlight the potential mechanisms behind a specific stage of Yoga Nidra (YN) practice, i.e. the formulation and repetition of Sankalpa…”. Regarding phrases like “first study results…” I tried to be more cautious (in this case I deleted “first”). However I had to consider also rev 2 asking me to clearly state in the abstract that studies cited are preliminary.
Comment 2: Introduction - The need for recovery and mental health in athletes is clearly presented.
Very up-to-date and comprehensive, with references to Yoga Nidra studies published in the last 3 years. The focus on the Sankalpa component within Yoga Nidra makes this article unique. However, how this contribution fills a gap in the literature could be more strongly emphasized.
Response 2: Thanks again for your positive comment. I also agree that how this contribution fills a gap in the literature should be emphasized. To this purpose I added the following sentence at the end of introduction section: “While mechanisms and effects of other exercises practiced during YN like breathing or body awareness exercises are documented [20], to the best of my knowledge, this is the first article that solicits new research lines focused on Sankalpa (and athletes) and tries to “unlock its power”.”
Comment 3: Methodology - The article is based on opinion and theoretical explanation, not empirical. The format is appropriate in this context. This section is not valid as it does not include analysis based on findings. However, the author has carefully interpreted the results of previous studies.
Response 3: Thanks for your comment on the section.
Comment 4: Findings-The potential impacts of Sankalpa are explained in a structured way.
Some suggestions are supported by examples (e.g. “I am calm”, “I express my sport potential”). However, the practical validity of the suggestions should be examined more critically. The reader should not be left with the question of what these suggestions are based on.
Response 4: Thank you for highlighting this point. Actually, Sankalpas suggested are purely examples. The most important thing is that they are formulated considering people/athletes’ own aspirations. For example, before or during competitions it could be useful formulate the intention to express the sport potential to the maximum extent possible, or stay quiet and calm after training to better recover and counterbalance stress. To better elucidate this I wrote: “Overall, athletes, always taking into account their personal aspirations, could use Sankalpa as an affirmation to overcome any weakness affecting their body, sport performance (and life in general), and to awaken any other strength they may feel is necessary to provide them with stress-recovery balance and mental health.
Immediately before or during competitions, mentally reciting something like "I am developing/I develop..." or "I am expressing my sport potential/I express," instead of, "I am going to develop..." or “I am going to express..” athletes could better create a Sankalpa that suits their needs. Other examples, especially when YN is executed after training and/or before competitions, could be “I am calm”, “I am successful…”
Comment 5: Discussion - Consistent and comprehensive with the relevant literature. However, the methodological limitations of research on the effects of Yoga Nidra and Sankalpa should be given more space. The limitations of non-evidence-based recommendations should be made clear, especially given the opinion piece format. There are some very good guidance (recommendations for qualitative/neurophysiological analysis, etc.). The potential of Sankalpa for mental healing and motivational boosting is well summarized.
Response 5: Thanks for your feedback on this section. In the revised manuscript I have now used more conditionality while I wrote a specific sentence in the conclusion section to remark methodological limitations of research concerning YN. It reads as follows: “Given the preliminary nature of YN research and the limited number of studies on the topic, the fact that mechanism/effects of Sankalpa herein described are merely potential should be remarked. On the other hand, grounded on the notion that Sankalpa can be analogous to positive self-instructions used to counteract dysfunctional cognition, it could be argued that establishing it might serve to initiate a process of change inside the subconscious mind…”
Comment 6: Conclusion - Concise and clear. However, in the conclusion, more cautious language (e.g. “Sankalpa may...”) should be preferred, in line with the nature of the opinion piece.
Response 6: Suggestion followed and, as stated also in response 5, in the revised manuscript I added a sentence in the conclusion to highlight that effects and mechanisms of Sankalpa are merely potential mainly because YN research is still limited. I also used more conditionality throughout the manuscript in line with the nature of this opinion manuscript.
Thank you once again for taking the time to review my manuscript. I sincerely hope my amendments will meet your approval.
Reviewer 4 Report
Comments and Suggestions for Authors
The abstract of the study is sufficient. The subject is clearly stated and explanations are provided.
The keywords are adequate.
Lines 28–30: The effects of yoga should be emphasized without referring to sleep quality. As it is currently written, the reader might assume that the study investigates sleep quality.
In the Introduction, the importance of recovery in athletes should be highlighted first. Following this, Yoga Nidra should be introduced as a recovery method.
More information should be provided on how Sankalpa reduces pre-competition stress and supports post-competition recovery.
An implementation suggestion is needed: How long before or after competition should it be practiced? Should it have a defined duration in minutes? How many days per week should it be applied?
The mechanism of Sankalpa should be explained.
Author Response
Dear Reviewer 4,
Comment 1: The abstract of the study is sufficient. The subject is clearly stated and explanations are provided. The keywords are adequate.
Response 1: Dear Reviewer, thanks for the positive feedback and also for the issues raised on my manuscript. Below, I responded to your queries point by point. To facilitate your work, any change was highlighted using yellow colour throughout the manuscript.
Comment 2: Lines 28–30 - The effects of yoga should be emphasized without referring to sleep quality. As it is currently written, the reader might assume that the study investigates sleep quality.
Response 2: Suggestion followed. The text reads now as follows: “Mental and physical recovery in competitive sports is one of the most crucial determinants of performance success and health [1]. To aid recovery and health, athletes should improve their recovery skills, adopting strategies that positively influence their autonomic nervous system and muscle tension. Relaxation strategies, such as mindfulness-based meditations, can be used for this purpose…”
Comment 3: In the Introduction, the importance of recovery in athletes should be highlighted first. Following this, Yoga Nidra should be introduced as a recovery method.
Response 3: Thank your for your feedback. Please, in this regard see my former response to your comment.
Comment 4: More information should be provided on how Sankalpa reduces pre-competition stress and supports post-competition recovery.
Response 4: Thanks for highlighting this point. For example, regarding recovery, I wrote: “YN and Sankalpa might have recently helped to increase karate athletes’ self-efficacy ratings, which are viewed as a relevant sport-specific aspect of recovery”. Or “In the early stages of YN practice, Sankalpa could act as a focal point for mental energies thereby concentrating and directing the mind towards a specific intention or goal. As the practice continues, the Sankalpa is repeated and this is believed to increase the energy necessary for manifesting intentions and accomplishing goals… In this way, Sankalpa could enable control over one’s own cognitive (e.g., feeling anxious) and physiological states (e.g., increased heart rate, muscle tension) but it could also strengthen awareness skills and create the space for inner harmony. The introspection and the conscious creation of a Sankalpa could lead athletes gain deeper insights into their values, aspirations, and needs (e.g., recovery needs), reinforcing their self-awareness. The more athletes become aware of their (recovery) needs, the higher satisfaction and contentment that underneath physical and mental health…”or “Overall, affirmations like Sankalpa may decrease stress by redirecting attention towards positive intensions, improve well-being and (sport) performance, and make people in general, and athletes in particular, prone to behavioral changes…”.
Comment 5: An implementation suggestion is needed: How long before or after competition should it be practiced? Should it have a defined duration in minutes? How many days per week should it be applied?
Response 5: Thank you for these important suggestions. In the revised version of the manuscript, in the section Sankalpa for Sport and Health, I specifically wrote: “Furthermore, athletes should be acknowledged that Sankalpa might be more effective when practiced within YN sessions due to calmness and receptiveness of the mind. Also, despite there are no time limits, few minutes dedicated to silently repeating Sankalpa within a longer practice could be sufficient. For example, during a YN session of about 30 minutes, 5 minutes might be dedicated to Sankalpa. Notably, a consistent practice is key. Importantly, YN should not be practiced too close to competitions to avoid excessive lowering of arousal levels due to the activation of the parasympathetic system [25]. Thus, the moments before competitions, reciting Sankalpa might be independent from the meditative practice. On the other hand, the day before competitions, or after trainings/competitions, athletes, according to their time constrains, can have a YN session where the intention is normally repeated at the beginning and at the end of the practice.”
Comment 6: The mechanism of Sankalpa should be explained.
Response 6: I understand your reasoning about this aspect. In this regard, please, consider what I added in the conclusion section: “ Given the preliminary nature of YN research and the limited number of studies on the topic, the fact that mechanism/effects of Sankalpa herein described are merely potential should be remarked. On the other hand, grounded on the notion that Sankalpa can be analogous to positive self-instructions used to counteract dysfunctional cognition, it could be argued that establishing it might serve to initiate a process of change inside the subconscious mind [4]. Also, in the section of potential mechanisms I wrote (for example): “As stated earlier, cognitive restructuring processes induced by Sankalpa might stop or reverse unhealthy thought patterns, resulting in personal accomplishment and greater mental health […]. However, Sankalpa could also be considered a remarkable motivational fuel. This motivation is essential to reach a state of “flow”, in which individuals are completely immersed in the activity they are doing, experiencing a sense of thorough satisfaction and fulfillment. This state is also associated with greater creativity, learning, and psychological well-being […], all aspects that could be enhanced through the regular practice of YN and Sankalpa”
However, only future research could better establish Sankalpa mechanisms and I hope that with this opinion paper the studies on the topic will grow accordingly. Thank you once again for the taking the time to revise my manuscript. I hope that my amendments will meet your approval.
Round 2
Reviewer 2 Report
Comments and Suggestions for Authors
Most of the reviewer's suggestions have been addressed. Thank you.